# Effectiveness of Nintendo Wii Fit© for Physical Therapy in Patients with Multiple Sclerosis: A Systematic Review and Meta-Analysis of Randomized Controlled Trials

**DOI:** 10.3390/jpm14090896

**Published:** 2024-08-24

**Authors:** Alvaro Alba-Rueda, Amaranta De Miguel-Rubio, David Lucena-Anton

**Affiliations:** 1Department of Nursing and Physiotherapy, University of Cadiz, 11009 Cadiz, Spain; alvaro.alba@uca.es (A.A.-R.); david.lucena@uca.es (D.L.-A.); 2Department of Nursing, Pharmacology and Physiotherapy, University of Cordoba, 14004 Cordoba, Spain; 3Biomedical Research and Innovation Institute of Cadiz (INiBICA), 11009 Cadiz, Spain

**Keywords:** multiple sclerosis, exergaming, videogames, exercise, physical therapy, neurological rehabilitation, systematic review, meta-analysis

## Abstract

Multiple sclerosis (MS) is a chronic, inflammatory, and autoimmune disease that mainly affects the central nervous system and currently has no cure. Exergaming is considered a non-immersive approach to improving functional and motor skills in the treatment of MS. The aim of this systematic review was to evaluate the effectiveness of the Nintendo Wii Fit© (NWF) on physical outcomes compared with control regimes in patients with MS. The search was performed in seven databases including articles published up to June 2024. The PICOS model was used to establish the study eligibility criteria. The Cochrane Collaboration tool and the PEDro scale were used to assess the risk of bias and evaluate the methodological quality of the studies, respectively. A meta-analysis using the standardized mean difference (SMD) and confidence interval (95% CI) was developed using the Review Manager 5.4 software. Seven articles were included in the systematic review. The statistical analysis showed favorable overall results for the NWF on functional mobility (SMD = 0.25; 95% CI = 0.09, 0.41) and fatigue (SMD = 0.41; 95% CI = 0.00, 0.82). In conclusion, this systematic review suggests that the NWF has shown favorable effects compared to control regimes on functional mobility and fatigue outcomes in patients with MS.

## 1. Introduction

Multiple sclerosis (MS) is a chronic, inflammatory, and autoimmune pathology that mainly affects the central nervous system, causing lesions such as demyelinating plaques and axonal damage that deteriorate neuronal conduction [1,2,3]. Genetics, environment, and lifestyle are predisposing risk factors for the diagnosis of this disease [4,5], which has a high prevalence worldwide (2.8 million people with MS (PwMS)) [6], especially in women (2:1) [7]. The diagnosis of MS is based on the 2017 last updated McDonald criteria [8]. MS usually appears in early adulthood (20–40 years), although it can present at any stage of life, having a great impact on the quality of life of PwMS due to a progressive increase in their disability [3,9].

The different forms of MS presentation are determined by the clinical course of the disease, ranging from relapsing-remitting MS (85% of diagnoses in PwMS), clinically isolated syndrome, radiologically isolated syndrome, and primary progressive MS to secondary progressive MS [10,11]. The clinical manifestations of the pathology are heterogeneous, depending on the affected area, and also follow a progressive evolution or take the form of exacerbation episodes [12]. MS signs and symptoms may be physical (fatigue, spasticity, strength deficit, pain, ataxia, proprioception impairment, or visual and balance problems) [13,14] or cognitive-emotional (depression or cognitive deficit) [15] and/or functional (gait deterioration, reduced manual dexterity, or sexual and urinary-rectal dysfunction) [13,16,17,18,19]. The treatment of MS must be considered from a multidisciplinary standpoint, with a combination of medical, pharmacological, physiotherapeutic, and occupational therapy care. Currently, it is a pathology with no cure, so treatment focuses on ameliorating symptoms and possible exacerbations to stabilize disease progression [20].

Regardless of disease severity, regular physical activity by PwMS has beneficial effects on functionality and overall health through neuroplastic changes in the brain [21,22]. The prescription of a combined training regime (resistance, endurance, aerobic, and functional and flexibility exercises) [21] is recommended for PwMS according to the Expanded Disability Status Scale score [23] and the individualized characteristics of each person [22,24]. An alternative tool for physical therapy treatment that has increased substantially in use over the last decade is virtual reality (VR) [25]. VR provides continuous visual and auditory feedback through a screen about the performance of the player, whose actions are collected by motion sensors and then simulated with an embodiment avatar created in a digital environment [26,27]. 

According to the level of immersion in this virtual world, exergames can correspond to non-immersive VR [28]. Exergames are an excellent way to encourage physical exercise at varying intensities in a playful way in populations with neurological diseases, both in the clinical setting and at home [26]. This approach also creates a safe, stimulating, and learning context in which to develop and repeat functional tasks similar to routine daily activities [29]. Moreover, the competitive and motivational aspects of this intervention improve adherence to treatment [30]. The Nintendo Wii© has been one of the most widely used devices in this research area since 2008 [31]. In this case, movements are detected in the majority of games via the device’s remote control (Nintendo Wii Remote© or Nintendo Wiimote©—Nintendo Co., Ltd., Kyoto, Japan), although some exergames need additional motion sensors, such as the Nintendo Wii Nunchuk©, used for two-handed tasks, or the Nintendo Wii Balance Board©, used as a force plate to monitor the center of gravity [32,33].

The existing literature provides evidence of the effectiveness of exergames in the rehabilitation of neurological pathologies [34]. In fact, several systematic reviews have demonstrated improvements in balance [35,36,37], physical functional capacity [36,37], and fatigue [36,37] for PwMS following this type of therapy. Furthermore, some systematic reviews with meta-analyses have considered Nintendo Wii© exergames as an effective neurorehabilitation treatment to improve upper limb function in cerebral palsy [38], functional mobility in Parkinson’s disease [39] and post-stroke patients [40], and balance in the pathologies already mentioned [41,42,43,44,45]. 

The Nintendo Wii Fit© (NWF), which features in most of the previous articles on the Nintendo Wii©, is a specific exergaming system complemented by the Nintendo Wii Balance Board©, with potential application as a rehabilitation tool in various clinical contexts [46]. This intervention provides dual-task training involving both motor and cognitive skills through minigames that enhance motor control, proprioception, and neuroplasticity, thereby reorganizing connections of the nervous system after injuries [47]. Besides this, the combination of moderate physical activity with interactive gaming offers positive effects on cardiovascular health, muscle strength, and metabolism [48].

The NWF is a highly beneficial and cost-effective option for MS rehabilitation, offering affordability, ease of use, motion-sensing technology, and interactive low-impact exercises that have been shown to significantly improve balance and coordination, as supported by high-quality research [39,40,41,42,43,44,45,46]. In addition, the use of a manual wireless controller and a force platform to interact with the digital landscape, graphical environment, and visual cue effects appears to offer greater benefits within the simpler and less distracting interface of the NWF compared to other systems, such as the Xbox Kinect™ [39]. In view of this information, we hypothesized that an intervention focused on the NWF could improve the physical outcomes of PwMS as it involves the practice of functional tasks in a fun and safe environment, and, consequently, helps the process of neuromotor rehabilitation in this disease. Therefore, the aim of this systematic review was to evaluate the effectiveness of the NWF on physical outcomes compared with control regimes in PwMS.

## 2. Materials and Methods

This systematic review and a meta-analysis of randomized controlled trials (RCTs) were carried out in line with the preferred reporting items for systematic reviews and meta-analyses (PRISMA) statement (Appendix A) [49]. The review protocol was registered and updated in the PROSPERO database (CRD42024531699).

### 2.1. Search Strategy

The literature search was conducted in PubMed, CENTRAL (Cochrane Central Register of Controlled Trials), CINAHL (Cumulative Index to Nursing & Allied Health Literature), Scopus, WoS (Web of Science), Medline, and PEDro (Physiotherapy Evidence Database) electronic databases, including articles published up to June 2024. Language or date filters were not applied. The following descriptor terms were used to create the search strategy: “Multiple sclerosis” [MeSH], “Disseminated sclerosis” [MeSH], “Nintendo Wii”, “Wii”, “Wii Fit”, “Wii Balance Board” and “exergaming” [MeSH]. A detailed search strategy with Boolean operators and the results for each database are presented in Appendix A. 

### 2.2. Eligibility Criteria

The inclusion criteria were defined by the PICOS model [50]: (1) population: adults with diagnosis of any MS disease types based on the McDonald criteria [8]; (2) intervention: experimental group (EG) using the NWF; (3) comparison: control group (CG) through conventional physical therapy, no treatment or usual care; (4) outcomes: functional mobility, fatigue and postural control; (5): study design: RCTs. Publications were excluded in the case of: (1) conference/congress communications or abstracts; (2) samples with combined neurological disorders (including PwMS) without separate results for outcomes in each population; (3) a PEDro scale score for methodological quality of <5 points. Based on the literature, only studies of high methodological quality, defined as a minimum PEDro score of 5 out of 10, were included in this review to ensure the validity of the results [51].

### 2.3. Study Selection Process and Data Extraction

Duplicated articles were removed after applying the previously mentioned search strategy in the scientific databases. Subsequently, the titles and abstracts were reviewed, and those publications that were not in line with the established eligibility criteria were excluded. The remaining articles were analyzed in depth, and the selected studies were included in the systematic review. Two independent reviewers (A.A.-R. and D.L.-A.) participated independently in the study selection process, analysis, data extraction, and risk of bias assessment. In case of dispute, a third reviewer (A.D.M.-R.) took part in resolving conflicts. The following data were extracted from each article: author/s, year, sample size, age, sex, MS form, Expanded Disability Status Scale score and the time since diagnosis of the participants, type of intervention, the frequency and duration of the intervention and of each session, study outcomes, measurement instruments, and the main results after the intervention (significant pre-post intragroup and intergroup differences).

### 2.4. Assessment of the Risk of Bias and Methodological Quality

The risk of bias in the studies was evaluated using version 2 of the Cochrane risk-of-bias tool for randomized trials (RoB 2) [52] and the Review Manager 5.4 software. A low, uncertain, or high risk was determined according to an evaluation of the different individualized items for each publication. Additionally, an overall summary of the risk of bias was determined, including the percentages of the previously assessed criteria.

The PEDro scale was used to assess the included RCTs as a valid measure of the methodological quality of clinical trials for physical therapy [53]. This scale includes items to evaluate external validity (criterion 1, which is not included in the final score), internal validity (criteria 2–9), and statistical information for interpreting the results (criteria 10 and 11) of clinical trials. According to the PEDro scale scores, studies were classified as being of fair (4–5), good (6–8), or excellent (9–10) quality [54]. 

### 2.5. Data Analysis

The meta-analysis was conducted using the Review Manager 5.4 software. First, the mean differences and standard deviations for each study group were calculated based on the post-intervention and baseline values, without considering the follow-up periods. This approach highlights the changes within each group without applying a specific cut-off point for evaluating the trial outcomes. Changes in effect size were analyzed by calculating the intergroup standardized mean difference (SMD) with a confidence interval (95% CI). An inverse variance method with continuous variables was used. Depending on the level of heterogeneity, either a fixed-effects (*p* > 0.05) or a random-effects (*p* < 0.05) model was used, along with the I^2^ statistic and the chi-squared test. I^2^ values between 0% and 40% indicate low heterogeneity; between 30% and 60% represent moderate heterogeneity; between 50% and 90% define substantial heterogeneity; and between 75% and 100% describe considerable heterogeneity [55]. The RCTs were grouped into different categories, based on the outcome measured, obtaining the overall results for the entire meta-analysis as well as the potential subtests within it.

## 3. Results

### 3.1. Study Selection

A total of 376 articles were initially identified during the study selection process. Out of these, 233 duplicate articles were removed. This left 143 articles, from which those not relevant to our search objectives, based on their titles and abstracts, were excluded. This resulted in 86 articles. After thoroughly applying the eligibility criteria, seven RCTs [56,57,58,59,60,61,62] were ultimately included in this systematic review and meta-analysis, as illustrated in Figure 1.

### 3.2. Description of the Studies

Regarding the articles included in the qualitative synthesis, a total of 288 participants (EG: *n* = 136, CG: *n* = 152; 76 men (26.4%) and 212 women (73.6%)) with MS were recruited. The sample size of the studies ranged from 16 [57] to 84 [58] participants. The mean age of the participants was 45.37 ± 6.02 years [56,57,58,59,60,61,62]. Relapsing-remitting was the most frequent form of MS [57,58,61,62]. The Expanded Disability Status Scale score of the participants was lower than 6.0 points [56,57,59,60,61,62], whereas the time since diagnosis varied between the onset of the disease and after more than 16 years [56,57,58,59,61,62].

In the experimental groups, the following NWF minigames were played: −Soccer Heading: The player must tilt on the balance board in the path of a soccer ball, dodging other flying objects [56,57,58,59,60,62].−Ski Slalom: The player must lean their body left or right to move the ski down a slalom course. In this activity, the Nintendo Wii Balance Board© is placed horizontally on the floor [56,57,58,59,60,62].−Table Tilt: The player must lean their body left, right, forward, and backward on the balance board to drop the balls into the holes [56,57,58,59,60,62].−Snowboard Slalom: The player must lean their body left or right to move the snowboard down a slalom course. In this activity, the Nintendo Wii Balance Board© is placed vertically on the floor [56,57,58].−Tightrope Walk: The player must walk on the balance board across the tightrope, bending their body and straightening their knees to jump when a black robotic jaw obstructs their path [56,57,58,59,60].−Zazen: The player must stay still on the balance board and concentrate on the candle until it completely burns down [56,59].−Penguin Slide: The player must tilt their body to left and right on the balance board to tip the iceberg and catch fish [57,58,59,62].−Ski Jump: The player must squat with their knees bent and push forward on the Nintendo Wii Balance Board© to gain speed. At the end of the ramp, the player must extend their knees and then keep their balance when landing [57].−Perfect 10: The player must sway their hips left, right, up, and down on the balance board for a score adding up to the total shown on the upper left of the screen [58].−Balance Bubble: The player must guide the avatar safely down a river, avoiding obstacles by leaning to the left, right, forward, and backward on the balance board [58,59,62].−Skateboard Arena: The player must propel a skateboard by removing one foot from the balance board and then placing it back, as on a real skateboard, while simultaneously leaning to the left and right to turn [58].−Rhythm Boxing: The player must punch the dartboards on the right or left (via the Nintendo Wiimote© and Nintendo Wii Nunchuk©), moving their feet at the same time on the balance board and following a rhythmic sequence [60].−Basic Step: The player must step on and off the Nintendo Wii Balance Board© in time with the on-screen steps [60].−Hula Hoop: The player must achieve as many spins of the hula hoop as possible in 70 s while on the balance board [60].−Torso Twist: The player must twist their body diagonally to both sides on the balance board, avoiding bending forward, to train their abdominal muscles [60].−Rowing Squat: The player must squat down on the Nintendo Wii Balance Board© while performing a rowing motion to train their thighs and back muscles [60].

The study by Thomas et al. [61] did not detail the NWF minigames used and included both Nintendo Wii Sports© and Nintendo Wii Sports Resorts© (both using software supplemented by Nintendo Wiimote© and Nintendo Wii Nunchuk© to develop different sport activities such as bowling, golf, baseball, or tennis) as alternative videogames in the Mii-vitaliSe program. In contrast, participants in the control groups did not receive any treatment [57,58,59,60,62] or continued to carry out their usual care [61]. The rest of the studies concerned conventional physical therapy (traditional balance training and strength exercises) [56,60]. 

The total duration of the interventions ranged from 4 weeks [56,60] to 6 months [61]. The frequency and the session time varied between a minimum of 1 session [61] to 4–5 sessions/week [59] and 30 min [58,59] to 60 min/session [56,60,62].

Multiple assessment tools were employed to evaluate the different outcomes of the participants. Functional mobility was assessed using the Berg balance scale [56,62], timed up-and-go test [58,61,62], timed 25-foot walk test [58,59], dynamic gait index [58], 12-item multiple sclerosis walking scale [58,60], timed chair stands test [58], activities-specific balance confidence scale [58], four-square step test [58,59], 2-min walk test [61], step test [61], 6-min walk test [62], and GAITRite [60]. Fatigue was evaluated using the modified fatigue impact scale [56], fatigue symptom inventory [61], and fatigue severity scale [62]. Postural control was measured by a force platform, which calculated the area [56] or the anterior-posterior and medial-lateral ranges [59,60] through eyes-open [56,59,60] or eyes-closed testing [56]; finally, the equilibrium quotient percentage score was determined using the sensory organization test [57] and the Equitest sensory organization test [61]. 

The results either showed significant differences within the same group before and after treatment [56,57,60,62], or between EG and CG [58,59]. Thomas et al. [61], representing the exception, used the mean differences adjusted for baseline. The level of significance was set at *p* ≤ 0.05.

The detailed characteristics of the articles included in this systematic review and meta-analysis are presented in Table 1.

The RCTs analyzed in this systematic review were of generally fair methodological quality (mean of the PEDro scale scores: 5.7 ± 0.95; range: 5–7). Conversely, 43% (3 out of 7) of the RCTs [58,59,61] demonstrated good methodological quality, with a score of ≥6 points. None of the RCTs met the criteria for the blinding of participants (C5) and for the blinded therapists administering the treatment (C6), as detailed in Table 2.

The assessment of the risk of bias of each RCT indicated that the study by Nilsagård et al. [58] showed the lowest risk (14%), whereas that by Robinson et al. [60] showed the highest risk (57%) (Figure 2).

Additionally, the overall summary of the risk of bias demonstrated that the lowest risk (0%) was found in the random sequence generation and selective reporting criteria, while the highest risk (100%) was presented by the blinding of participants and personnel (Figure 3).

### 3.3. Synthesis of Meta-Analysis Results between Experimental and Control Groups

A total of 86% (6 out of 7) of the RCTs [56,58,59,60,61,62] were included in the quantitative synthesis. The mean, standard deviation, and sample sizes of both the experimental and control group of each RCT, in addition to the meta-analysis results with the SMD (95% CI), are presented according to the outcome analyzed. 

#### 3.3.1. Functional Mobility

A meta-analysis was performed for the functional mobility outcome, including 86% (6 out of 7) of the RCTs [56,58,59,60,61,62]. The functional mobility results showed a low degree of heterogeneity (*p* = 0.48; I^2^ = 0%). This outcome was divided into different subtests. The overall results of the subtests’ meta-analyses were favorable for the experimental group for the Berg balance scale (SMD = 0.83; 95% CI = 0.32, 1.33) and the timed up-and-go test (SMD = 0.37; 95% CI = 0.04, 0.70). The four-square step test (SMD = 0.02; 95% CI = −0.34, 0.38), timed 25-foot walk (SMD = 0.15; 95% CI = −0.20, 0.51), and 12-item multiple sclerosis walking scale (SMD = 0.17; 95% CI = −0.15, 0.48) did not report significant results. The overall result of the entire meta-analysis was conclusively in favor of the experimental group (SMD = 0.25; 95% CI = 0.09, 0.41). The forest plots are shown in Figure 4.

#### 3.3.2. Fatigue

A meta-analysis was performed for the fatigue outcome, including 43% (3 out of 7) of the RCTs [56,61,62]. A low degree of heterogeneity (*p* = 0.29; I^2^ = 19%) for this outcome was found. The overall result of the entire meta-analysis was favorable for the experimental group (SMD = 0.41; 95% CI = 0.00, 0.82). The forest plots are shown in Figure 5.

## 4. Discussion

This study aimed to evaluate the effectiveness of the NWF on physical outcomes in PwMS. A total of seven articles, involving 288 participants, were included in this systematic review and meta-analysis. To the best of our knowledge, this is the first meta-analysis of RCTs on this topic. The meta-analysis revealed significant improvements in functional mobility and fatigue, concluding that NWF usage is more effective than the control regimes for enhancing physical outcomes in PwMS.

In this sense, our results are consistent with the existing literature. In relation to functional mobility, previous meta-analyses on exergaming have shown significant improvements in scores for the Berg balance scale [35,36] and 6-min walk test [36], and shown favorable effects on the timed up-and-go test [36] and the 10-m walking test [36] for PwMS compared to conventional rehabilitation. Studies on NWF usage also indicate benefits for other conditions when compared to exercise interventions [40,43,44,45,63] or no treatment [64], including improvements on the Berg balance scale [40,43,44,45,63,64], timed up-and-go test [40,43,44,63,64], functional reach test [43,44], 30-s chair stand test [64], and functional independence measure [40] for populations such as Parkinson’s patients, stroke victims, and older adults [40,43,44,63,64]. However, a study by Marotta et al. [39] reported non-significant effects on functional locomotion when comparing the NWF with the Kinect Xbox system. Additionally, increasing the number of RCTs comparing the efficacy of two specific exergaming systems across various variables, as demonstrated by Yazgan et al. [62], would enhance the quality of systematic reviews in this field. This approach would provide valuable information on the optimal exergaming intervention, based on the primary symptoms and characteristics of each specific pathology.

This meta-analysis also reported improvements in fatigue. Nevertheless, because of the lack of meta-analyses analyzing the effects of NWF intervention on this outcome, the published literature has not involved any modality of exergame training. A previous meta-analysis by Elhusein et al. [36] showed conclusive favorable results for an exergaming intervention compared to conventional physical therapy on the modified fatigue impact scale for PwMS. Under different conditions, the study by Wu et al. [65] reported significant differences in fatigue severity between exergames compared with non-exercise in female patients with fibromyalgia. As illustrated in Figure 5, the diversity of measurement instruments used to assess fatigue prevented the formation of subgroups in the meta-analysis. Subgroup analyses are a crucial element of meta-analyses, as their absence implies a barrier to determining whether the pooled effect sizes in these subgroups differ significantly [66]. Consequently, a larger number of RCTs evaluating fatigue is needed to achieve significant results for each scale used to measure this outcome.

Although a meta-analysis on postural control was not feasible due to the heterogeneous presentation of data, significant intragroup differences were observed following NWF intervention in conditions, regarding performance in eyes-open tests [56,60], eyes-closed tests [56], and in certain subtests of the sensory organization test [57] and the Equitest sensory organization test [61]. Additionally, Prosperini et al. [59] demonstrated the positive effects of NWF usage on eyes-open postural control compared to no treatment. Furthermore, integrating a visual feedback component into balance training may support central nervous system vestibular compensation following peripheral labyrinthine disorders that impair postural control [67].

Beyond such physical improvements, the potential benefits related to quality of life, as identified in the selected RCTs, are also worth discussing. The NWF intervention revealed significant differences between groups, compared to no treatment, in reducing the impact of MS on quality of life, as assessed by the 29-item MS impact scale (*p* = 0.023) [59]. Additionally, significant differences within the NWF group were observed using the multiple sclerosis international quality of life questionnaire (*p* = 0.001) [62]. These findings suggest that integrating the NWF into MS treatment can enhance patients’ quality of life by improving physical outcomes, reducing stress, and providing an accessible and enjoyable form of exercise that benefits both physical and mental well-being [68].

In terms of the characteristics of the included studies, the predominance of relapsing-remitting MS and female sex aligns with the typical clinical presentation of MS disease [7]. Notwithstanding, variations in participant age, time since diagnosis, and expanded disability status scale scores suggest differing functional levels and treatment objectives [69], highlighting the potential benefits of stratifying samples in future RCTs. On the other hand, the diversity in intervention duration, frequency, and session duration could complicate the detection of changes within or between groups if NWF intervention does not adhere to combined-training recommendations for PwMS [24]. Moreover, the sample sizes ranged from 30 to 84 participants (with the exception of the study by Lee [57]); despite not being pilot trials, the results require cautious interpretation due to possible convenience sampling, meaning that the results may not accurately represent the general MS population [70]. We would also like to point out that the functional mobility outcome includes both static and dynamic balance assessments, as these aspects play a crucial role in the performance of lower-limb functional movements [71].

Due to the significant findings related to functional mobility and fatigue, we support integrating the NWF system into therapy centers and telerehabilitation as a practical, feasible, well-received, and enjoyable intervention [26]. Clinicians can incorporate the NWF into regular therapy sessions by providing tailored and supervised fall prevention training [46]. In addition, educating patients on the safe use of the program at home promotes independent rehabilitation, with remote support enhancing their autonomy and emphasizing the intervention’s flexibility to ensure continuous care and long-term commitment [72]. Furthermore, incorporating the NWF into rehabilitation programs could significantly reduce healthcare expenses. Its cost-effectiveness relative to traditional rehabilitation tools, along with its capability for efficient remote therapy, has the potential to decrease overall spending on physical therapy services [73].

Specifically, for PwMS, the NWF could be integrated into physical therapy protocols as a valuable complement to traditional training, particularly during relapses [36]. Engaging in home-based exercise with 10–40 min of NWF training two or more days per week, depending on the disability level, may provide a practical, effective, and safe method for managing MS symptoms [24,74]. Additionally, there is a need to create user-centered exergames that feature tailored audio-visual designs, varying difficulty levels, and adaptations for physical and cognitive impairments [75]. Consequently, physical therapists should consider each patient’s background, health profile, and preferences when designing personalized exercise programs using the NWF software for this population [76]. 

While this systematic review and the meta-analysis provide insights into the potential benefits of exergaming for improving physical outcomes in PwMS, several limitations need to be considered. First, the use of SMD instead of mean differences, due to heterogeneity in measurement instruments, reduced the statistical power of the meta-analyses, warranting careful interpretation of the findings. Second, the included RCTs reported a high risk of performance and detection bias because of the lack of blinding among participants, therapists, and, sometimes, assessors. Moreover, larger sample sizes, long-term follow-up assessments to evaluate the sustained effects of NWF interventions and more homogeneous participant characteristics are needed to establish robust conclusions and generalize the results. Finally, the RCTs did not include objective outcomes based on the International Classification of Functioning, Disability, and Health for MS conditions [77].

However, despite these restrictions, this article provides preliminary evidence regarding the effect of the NWF system on physical outcomes for PwMS. Therefore, further well-designed RCTs using rigorous methodologies are essential to guide future research focused on developing and integrating exergaming programs in healthcare.

## 5. Conclusions

In view of these potential findings, this systematic review and meta-analysis suggest that NWF intervention is more effective than control regimes (usual care, no intervention, or conventional physical therapy) in PwMS. In particular, NWF usage showed significant improvements in functional mobility and fatigue outcomes in this population. Due to its accessible nature and the ability to customize routine activities in a safe virtual environment, the NWF could serve as a valuable therapeutic alternative for physical therapy, enhancing the motivation and engagement of PwMS in both clinical settings and home-based neurorehabilitation. Nevertheless, further research based on this specific approach is required to establish solid conclusions.

## Figures and Tables

**Figure 1 jpm-14-00896-f001:**
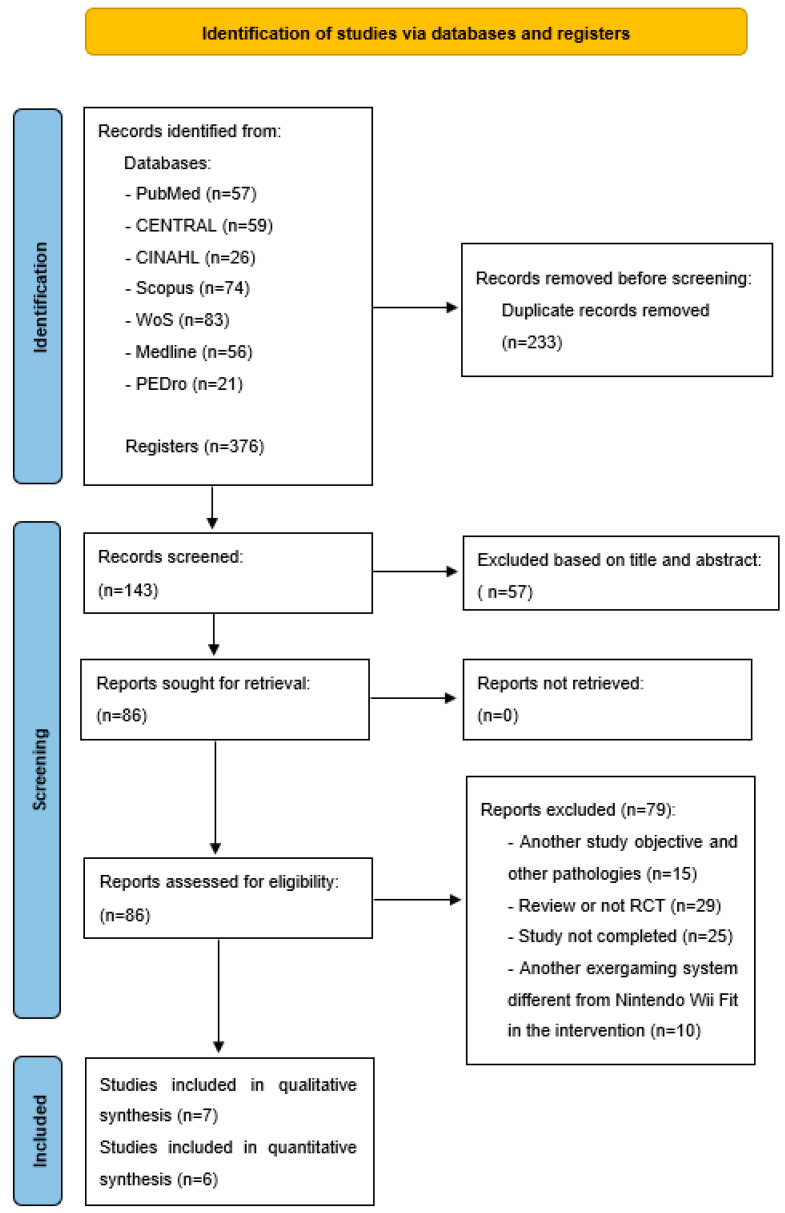
Flow diagram of the systematic review and meta-analysis following the PRISMA (preferred reporting items for systematic reviews and meta-analyses) recommendations [49].

**Figure 2 jpm-14-00896-f002:**
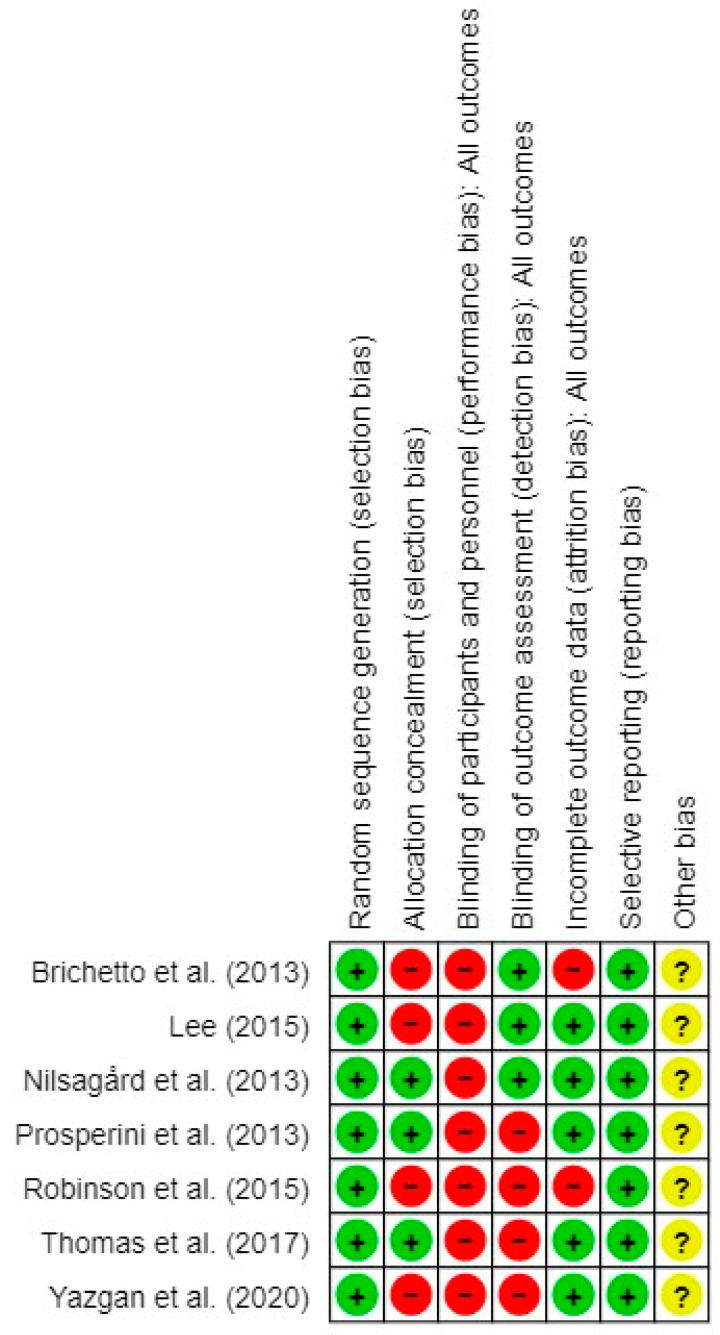
Risk of bias in the randomized controlled trials included in the systematic review and meta-analysis [56,57,58,59,60,61,62]. The green circle (+) indicates a low risk of bias, the yellow circle (?) describes an unclear risk of bias, and the red circle (−) indicates a high risk of bias.

**Figure 3 jpm-14-00896-f003:**
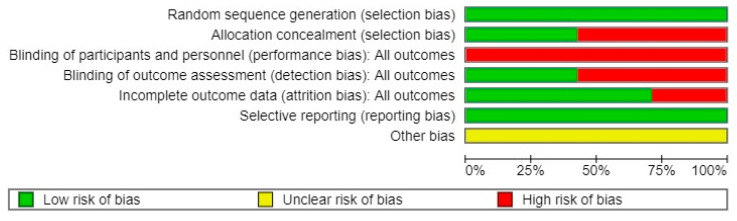
Overall risk of bias, with each category presented as a percentage.

**Figure 4 jpm-14-00896-f004:**
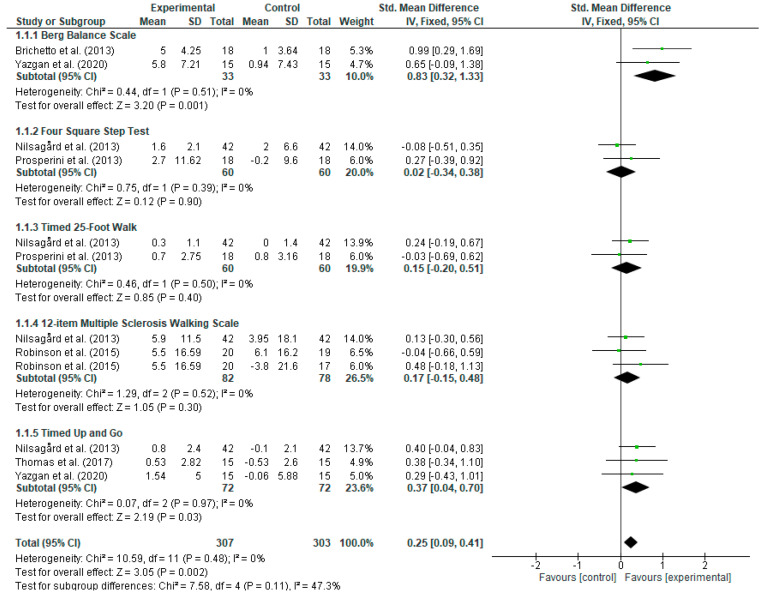
Forest plot for functional mobility [56,58,59,60,61,62]. CI: Confidence interval; IV: inverse variance; SD: standard deviation.

**Figure 5 jpm-14-00896-f005:**
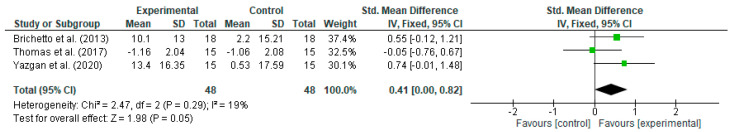
Forest plot for fatigue [56,61,62]. CI: Confidence interval; IV: inverse variance; SD: standard deviation.

**Table 1 jpm-14-00896-t001:** Main characteristics of the randomized controlled trials included in the systematic review and meta-analysis [53,54,55,56,57,58,59].

	Participants			Intervention					
Author/s (Year)	Sample Size (Male/ Female)	MS Form EDSS (Points) Time since Diagnosis (Years)	EG/CG: n (Age ± SD)	EG	CG	Duration and Frequency	Outcome	Measuring Instrument	Results
Brichetto et al. (2013) [56]	36 (14/22)	EG: ND MS form 3.9 ± 1.6 points 11.2 ± 6.4 years CG: ND MS form 4.3 ± 1.6 points 12.3 ± 7.2 years	EG: 18 (40.7 ± 11.5 y) CG: 18 (43.2 ± 10.6 y)	NWF: Soccer Heading, Ski Slalom, Table Tilt, Snowboard Slalom, Tightrope Walk, and Zazen	CPT: Static and dynamic ex (single or double leg) + equilibrium board + half-kneeling ex.	4 weeks 3 sessions /week 60 min /session	(1) Functional mobility (2) Postural control (3) Fatigue	(1) BBS (2) Force Platform (3) MFIS	SID: (1) EG: *p* < 0.05; CG: *p* < 0.05 (2) Eyes-Open FP → EG: *p* < 0.05; CG: *p* < 0.05 Eyes-Closed FP → EG: *p* < 0.05; CG: *p* < 0.05 (3) EG: *p* < 0.05
Lee (2015) [57]	16 (6/10)	EG: 2 PP 5 RR 1 SP 3.0–5.0 points 9.52 ± 5.2 years CG: 1 PP 5 RR 2 SP 3.0–5.0 points 10.12 ± 5.7 years	EG: 8 (39.2 ± 7.2 y) CG: 8 (41.5 ± 8.3 y)	NWF: Soccer Heading, Table Tilt, Penguin Slide, Ski Slalom, Snowboard Slalom, Tightrope Walk, and Ski Jump	No treatment	8 weeks 3 sessions /week 40 min /session	(1) Postural control	(1) SOT (Force Platform)	SID: (1) SOT-5 → EG: *p* < 0.05 SOT-6 → EG: *p* < 0.05 VES → EG: *p* < 0.05
Nilsagård et al. (2013) [58]	84 (20/64)	EG: 3 PP 26 RR 13 SP ND EDSS score 12.5 ± 8.0 years CG: 1 PP 28 RR 13 SP ND EDSS score 12.2 ± 9.2 years	EG: 42 (50.0 ± 11.5 y) CG: 42 (49.4 ± 11.1 y)	NWF: Penguin Slide, Ski Slalom, Snowboard Slalom, Perfect 10, Soccer Heading, Table Tilt, Tightrope Walk, Balance Bubble, and Skateboard Arena	No treatment	6–7 weeks 2 sessions /week 30 min /session	(1) Functional mobility	(1) TUG, T25-FW, DGI, MSWS-12, TCS, ABC, FSST	No SDBG (EG/CG): *p* > 0.05
Thomas et al. (2017) [61]	30 (3/27)	EG: 12 RR 3 SP 1.0–6.0 points <1–>16 years CG: 1 PP 9 RR 2 SP 3 O 1.0–6.0 points <1–>16 years	EG: 15 (50.9 ± 8.08 y) CG: 15 (47.6 ± 9.26 y)	Mii-vitaliSe program: NWF (ND Minigames) + Wii Sports + Wii Sports Resort	Usual care: Multidisciplinary support	6 months ≥1 session /week ND minutes /session	(1) Fatigue (2) Functional mobility (3) Postural control	(1) FSI (2) 2-MWT, TUG, ST (3) ESOT (Force Platform)	MDAB (95% CI): (1) 0.06 (−1.26, 1.38) (2) 2-MWT → 1.13 (−19.61, 21.86) TUG → −1.06 (−2.70, 0.58) ST → 2.61 (0.03, 5.18) (3) ESOT-1 → 2.00 (−2.96, 6.97); ESOT-2 → 5.62 (−3.79, 15.04); ESOT-3 → 4.41 (−3.87, 12.69); ESOT-4 → 7.91 (0.57, 15.24); ESOT-5 → 0.41 (−14.25, 15.07)
Yazgan et al. (2020) [62]	30 (4/26)	EG:1 PP 11 RR 1 SP 2 O 4.16 ± 1.37 points 12.06 ± 6.56 years CG: 14 RR 1 O 4.06 ± 1.26 points 11.06 ± 5.70 years	EG: 15 (47.46 ± 10.53 y) CG: 15 (40.66 ± 8.82 y)	NWF: Penguin Slide, Table Tilt, Ski Slalom, Soccer Heading, and Balance Bubble	No treatment	8 weeks 2 sessions /week 60 min /session	(1) Functional mobility (2) Fatigue	(1) BBS, TUG, 6-MWT (2) FSS	SID: (1) EG: *p* = 0.001; CG: *p* = 0.028 TUG → EG: *p* = 0.003 6-MWT → EG: *p* = 0.001 (2) EG: *p* = 0.002
Prosperini et al. (2013) [59]	36 (11/25)	EG: RR and SP (ND) 1.5–5.0 points 12.2 ± 6.0 years CG: RR and SP (ND) 1.5–5.0 points 9.3 ± 5.3 years	EG: 18 (35.3 ± 8.6 y) CG: 18 (37.1 ± 8.8 y)	NWF: Zazen, Table Tilt, Ski Slalom, Penguin Slide, Tightrope Walk, Soccer Heading, and Balance Bubble	No treatment	12 weeks 4–5 sessions /week 30 min /session	(1) Postural control (2) Functional mobility	(1) Force Platform (2) FSST, T25-FW	SDBG (EG/CG): (1) Eyes-Open FP → *p* = 0.016 (2) FSST → *p* = 0.034, 25-FWT → *p* = 0.048
Robinson et al. (2015) [60]	56 (18/38)	EG: ND MS form <6.0 points ND years CG1: ND MS form <6.0 points ND years CG2: ND MS form <6.0 points ND years	EG: 20 (52.6 ± 6.1 y) CG1: 19 (53.9 ± 6.5 y) CG2: 17 (51.9 ± 4.7 y)	NWF: Soccer Heading, Ski Slalom, Table Tilt, Tightrope Walk, Rhythm Boxing, Basic Step, Hula Hoop, Torso Twist, and Rowing Squat	CG1: CPT: Traditional balance training CG2: No treatment	4 weeks 2 sessions /week 40–60 min /session	(1) Postural control (2) Functional mobility	(1) Force Platform (2) MSWS-12, GAITRite	SID: (1) Eyes-open AP Range → EG: *p* = 0.04; CG1: *p* = 0.04 Eyes- open ML Range → EG: *p* = 0.04; CG1: *p* = 0.01 Eyes-open CoP Velocity → EG: *p* = 0.01 (2) MSWS-12 → CG1: *p* = 0.03

2-MWT: 2-min walk test; 6-MWT: 6-min walk test; 9-HPT: 9-hole peg test; ABC: activities-specific balance confidence scale; AP: anterior-posterior; BBS: Berg balance scale; CG: control group; CI: confidence interval; CoP: center of pressure; CPT: conventional physical therapy; DGI: dynamic gait index; EDSS: expanded disability status scale; EG: experimental group; ESOT: Equitest sensory organization test; FSI: fatigue symptom inventory; FSS: fatigue severity scale; FSST: four-square step test; MDAB: mean differences adjusted for baseline; MFIS: modified fatigue impact scale; ML: medial–lateral; MS: multiple sclerosis; MSWS-12: 12-item multiple sclerosis walking scale; ND: not described; NWF: Nintendo Wii Fit; O: others; PP: primary progressive; RR: relapsing–remitting; SD: standard deviation; SDBG: significant differences between groups; SID: significant intragroup differences; SOT: sensory organization test; SP: secondary progressive; ST: step test; T25-FW: timed 25-foot walk test; TCS: timed chair stands test; TUG: timed up-and-go test; VES: vestibular ratio 3.3. Risk of bias and methodological quality.

**Table 2 jpm-14-00896-t002:** Methodological quality results of the randomized controlled trials included in the systematic review and meta-analysis, judged according to the PEDro scale [56,57,58,59,60,61,62].

Author/s (Year)	C1*	C2	C3	C4	C5	C6	C7	C8	C9	C10	C11	Total Score (out of 10)	Methodological Quality
Brichetto et al. (2013) [56]	1	1	0	1	0	0	1	0	0	1	1	5	Fair
Lee (2015) [57]	1	1	0	1	0	0	1	1	0	0	1	5	Fair
Nilsagård et al. (2013) [58]	1	1	1	1	0	0	1	1	0	1	1	7	Good
Prosperini et al. (2013) [59]	1	1	1	1	0	0	0	1	0	1	1	6	Good
Robinson et al. (2015) [60]	1	1	0	1	0	0	0	0	1	1	1	5	Fair
Thomas et al. (2017) [61]	1	1	1	1	0	0	0	1	1	1	1	7	Good
Yazgan et al. (2020) [62]	1	1	0	1	0	0	0	1	0	1	1	5	Fair

Range: 0–10. C1* is not used to calculate the PEDro score. Note: “1” indicates that a study meets that criterion, and “0” means that the study does not meet the criterion or does not provide sufficient information to ensure it. C1* = the choice criteria have been specified. C2 = participants were randomly assigned to groups. C3 = treatment assignment was performed in a concealed manner. C4 = groups had similar characteristics at baseline. C5 = blinding of participants. C6 = blinded therapists administered the treatment. C7 = blindness of assessors collecting measurements. C8 = measures of at least one of the key outcomes were obtained from >85% of the subjects initially assigned to the groups. C9 = results were presented for all subjects who received treatment or were assigned to the control group; when this could not be achieved, data for at least one key outcome were analyzed on an “intention-to-treat” basis. C10 = results of statistical comparisons between groups were reported for at least one key outcome. C11 = the study provides point and variability measures for at least one key outcome.

## Data Availability

The datasets analyzed for this study can be found in the manuscript and Appendix A.

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
