# Peer review of "Effectiveness of Nintendo Wii Fit© for Physical Therapy in Patients with Multiple Sclerosis: A Systematic Review and Meta-Analysis of Randomized Controlled Trials"

_jpm, 2024, doi:10.3390/jpm14090896_

Round 1

Reviewer 1 Report

Comments and Suggestions for Authors

The systemic review titled "Effectiveness of Nintendo Wii Fit© for Physical Therapy in Patients with Multiple Sclerosis: A Systematic Review and Meta-Analysis of Randomized Controlled Trials" highlights the effectiveness of Nintendo Wii Fit for physical therapy in patients with Multiple Sclerosis. This study would significantly contribute to neurorehabilitation and establish the use of NWF in physical therapy for MS patients. I have some minor recommendations for the authors that would help enhance the overall quality of the manuscript:

1- Discuss the potential benefits of NWF in enhancing patient's quality of life and healthcare costs.

2- Discuss the practical implications of this study for clinicians NWF and how NWF can be integrated into physical therapy protocols for MS patients?

3- Elaborate more regarding the physiological and neurological pathways/mechanisms involved in NWF.

4- Justify the reasons for choosing a score of less than 5 as the cut-off on the PEDro score.

5- Proofread the whole manuscript once to avoid minor grammatical errors.

Comments on the Quality of English Language

The English language is fine. 

Author Response

We sincerely appreciate the valuable feedback on our manuscript titled “Effectiveness of Nintendo Wii Fit© for Physical Therapy in Patients with Multiple Sclerosis: A Systematic Review and Meta-Analysis of Randomized Controlled Trials” (ID: jpm-3145809). We highly appreciate the suggestions, as they significantly enhance the quality of our work. Below, we have provided a detailed, point-by-point response to each of your recommendations:

Point 1: Discuss the potential benefits of NWF in enhancing patient's quality of life and healthcare costs.

Response 1: We agree with this comment. Therefore, we have included in the Discussion section new references and relevant considerations related to the potential benefits of NWF in enhancing patient's quality of life (Lines 408-416) and healthcare costs (Lines 438-441).

Point 2: Discuss the practical implications of this study for clinicians NWF and how NWF can be integrated into physical therapy protocols for MS patients?

Response 2: We agree with this comment. Therefore, we have included in the Discussion section new references and relevant considerations related to the practical implications of this NWF study for clinicians (Lines 431-438) and how NWF can be integrated into physical therapy protocols for MS patients (Lines 442-450).

Point 3: Elaborate more regarding the physiological and neurological pathways/mechanisms involved in NWF.

Response 3: We agree with this comment. Therefore, we have included in the Introduction section new references and relevant considerations related to the physiological and neurological pathways/mechanisms involved in NWF (Lines 85-89).

Point 4: Justify the reasons for choosing a score of less than 5 as the cut-off on the PEDro score.

Response 4: We agree with this comment. Therefore, we have included in the Materials and Methods section a new reference to justify the punctuation of less than 5 as the cut-off on the PEDro score (Lines 127-129).

Point 5: Proofread the whole manuscript once to avoid minor grammatical errors.

Response 5: Please see the attachment (Certificate of Proof-Reading-Service of this manuscript).

Reviewer 2 Report

Comments and Suggestions for Authors

This is an interesting meta-analysis study exploring the effectiveness of NWF in improving functional mobility and fatigue outcomes in MS patients. Although the work is of interest, the reviewer is not convinced that the findings presented have the potential significance that we require for publication in JPM. Overall, the paper is more suitable for specialty journal and more work is required for a broader readership audience.

Author Response

We sincerely appreciate the valuable feedback on our manuscript titled “Effectiveness of Nintendo Wii Fit© for Physical Therapy in Patients with Multiple Sclerosis: A Systematic Review and Meta-Analysis of Randomized Controlled Trials” (ID: jpm-3145809).

Point 1: Although the work is of interest, the reviewer is not convinced that the findings presented have the potential significance that we require for publication in JPM. Overall, the paper is more suitable for specialty journal and more work is required for a broader readership audience.

Response 1: Thank you for your feedback and for recognizing the interest in our work. We understand your concerns regarding the potential significance of our findings. We believe that the implications of our research could contribute to a broader understanding within the field of using technologies in neurorehabilitation. Specifically, this systematic review aimed to assess the effectiveness of Nintendo Wii Fit© on physical outcomes in patients with multiple sclerosis compared to control regimes. Seven studies were analyzed, using standardized mean difference and confidence intervals (95%). The results showed that Nintendo Wii Fit© had a positive impact on functional mobility and fatigue in patients with multiple sclerosis, suggesting that Nintendo Wii Fit© may be beneficial in improving these outcomes. The paper has been submitted to the Special Issue: “Advances in Emerging Technologies for Rehabilitation: Personalized Perspective” of Journal of Personalized Medicine. This Special Issue (https://www.mdpi.com/journal/jpm/special_issues/RC4S5GVM92) aims to cover the following topics that need to be explored: personalized treatment and management, use of new technologies in rehabilitation. Therefore, we consider that the paper aligns wih these topics and objectives and it could a valuable contribution to the technology-supported field.

Reviewer 3 Report

Comments and Suggestions for Authors

Comments:

This is a systematic review and meta-analysis of RCT evaluating the effectiveness of Nintendo Wii Fit for physical therapy in patients with Multiple sclerosis. The authors have included 7 RCTs in the review and report that NWF had shown a favourable effect on functional mobility and fatigue outcomes in patients with MS. The research question is clinically relevant. The study methodology, analysis and their interpretations are appropriate. Th manuscript is well structured and easy to read. The authors need to address the following points:

1-       Authors need to justify why they intended to limit their research question to only one specific exercise game i.e Nintendo Wii, while other exercise games have been used in rehabilitation in MS patients. The justification of the same may be provided in the introduction.

2-       The inclusion criteria should mention which type of MS patients were included in the study?

3-       For Risk of Bias assessment, ROB 2 tool should have been used.

4-       What was the time cut off considered for the outcome assessments in this review? The included studies have different outcome periods.

5-       In the baseline characteristics table, the disease modifying agents used in different groups should be provided.

Comments on the Quality of English Language

Minor editing of the English language is required 

Author Response

We sincerely appreciate the valuable feedback on our manuscript titled “Effectiveness of Nintendo Wii Fit© for Physical Therapy in Patients with Multiple Sclerosis: A Systematic Review and Meta-Analysis of Randomized Controlled Trials” (ID: jpm-3145809). We highly appreciate the suggestions, as they significantly enhance the quality of our work. Below, we have provided a detailed, point-by-point response to each of your recommendations:

Point 1: Authors need to justify why they intended to limit their research question to only one specific exercise game i.e Nintendo Wii, while other exercise games have been used in rehabilitation in MS patients. The justification of the same may be provided in the introduction.

Response 1: We agree with this comment. Therefore, we have included in the Introduction section new relevant considerations to justify why we limited the research question to a single specific exercise game (NWF) (Lines 90-96).

Point 2: The inclusion criteria should mention which type of MS patients were included in the study?

Response 2: Yes, we agree with this comment. Therefore, we have revised the Materials and Methods section to include the types of MS in the population (P) inclusion criteria (Lines 119-120).

Point 3: For Risk of Bias assessment, ROB 2 tool should have been used.

Response 3: Yes, we agree with this comment. Therefore, we have revised the Materials and Methods section to include a new reference for the ROB 2 tool (Lines 144-145) and updated the Results section to modify Figures 2 and 3 (please see the revised manuscript).

Point 4: What was the time cut off considered for the outcome assessments in this review? The included studies have different outcome periods.

Response 4: We agree with this suggestion. To clarify the issue, we have included explanatory sentences on Lines 156-160: “First, the mean differences and standard deviations for each study group were calculated based on the post-intervention and baseline values without considering the follow-up periods. This approach highlights the changes within each group without applying a specific cut-off point for evaluating the trial outcomes.”.

Point 5: In the baseline characteristics table, the disease modifying agents used in different groups should be provided.

Response 5: We agree with this suggestion. We had originally intended to include such information in the systematic review, but it is not provided by the included studies.

Round 2

Reviewer 2 Report

Comments and Suggestions for Authors

The reviewer is impressed by the revised version and has no further concern.